# Overdue Calcium Oscillation Causes Polyspermy but Possibly Permits Normal Development in Mouse Eggs

**DOI:** 10.3390/ijms25010285

**Published:** 2023-12-24

**Authors:** Mio Fukuoka, Woojin Kang, Daiki Katano, Sae Horiike, Mami Miyado, Mamoru Tanaka, Kenji Miyado, Mitsutoshi Yamada

**Affiliations:** 1Department of Obstetrics and Gynecology, Keio University School of Medicine, 35 Shinanomachi, Shinjuku-ku, Tokyo 160-8582, Japan; mio04aa@keio.jp (M.F.); mtanaka@keio.jp (M.T.); mitutosi@a6.keio.jp (M.Y.); 2Department of Reproductive Biology, National Research Institute for Child Health and Development, 2-10-1 Okura, Setagaya-ku, Tokyo 157-8535, Japan; kwjbear@gmail.com (W.K.); ktn990419@gmail.com (D.K.); 11622035@nodai.ac.jp (S.H.); miyado@nm.beppu-u.ac.jp (M.M.)

**Keywords:** Ca^2+^ oscillation, citrate synthase, polyspermy, normal development

## Abstract

In some non-mammalian eggs, the fusion of one egg and multiple sperm (polyspermy) induces a robust rise in intracellular calcium ion (Ca^2+^) concentration due to a shortage of inducers carried by a single sperm. Instead, one of the sperm nuclei is selected inside the egg for normal embryogenesis. Polyspermy also occurs during the in vitro fertilization of human eggs; however, the fate of such eggs is still under debate. Hence, the relationship between polyspermy and repetitive Ca^2+^ increases (Ca^2+^ oscillation) in mammals remains unknown. To address this issue, we used mouse sperm lacking extramitochondrial citrate synthase (eCS), which functions as a Ca^2+^ oscillation inducer; its lack causes retarded Ca^2+^ oscillation initiation (*eCs*-KO sperm). Elevated sperm concentrations normalize Ca^2+^ oscillation initiation. As expected, eCS deficiency enhanced polyspermy in both zona pellucida (ZP)-free and ZP-intact eggs despite producing the next generation of *eCs*-KO males. In conclusion, similarly to non-mammalian eggs, mouse eggs may develop normally under polyspermy conditions caused by problematic Ca^2+^ oscillation.

## 1. Introduction

Monospermic fertilization (monospermy), which is generally considered successful, is ensured by preventing extra sperm fusion. Polyspermic fertilization (polyspermy) is believed to be a pathological phenomenon in mammals and, more specifically, in humans [1,2,3,4]. Alternatively, a single calcium (Ca^2+^) rise, stimulated by polyspermy, induces egg activation in most insects and some amphibians [5,6]. “One sperm nucleus” is selected in their egg cytoplasm to proceed with the normal two-cell formation and subsequent embryogenesis [5,6]. In these organisms, polyspermy means “physiological polyspermy”, leading to “cytoplasmic sperm selection”. However, the physiological roles of polyspermy in mammals remain unclear.

In the mitochondria, citrate synthase (CS) catalyzes citrate production as the first reaction of the tricarboxylic acid (TCA) cycle (also known as the citric acid or Krebs cycle). A single gene produces variants corresponding to CS and extramitochondrial CS (eCS) lacking a mitochondrial-targeting sequence in various species of plants and animals [7]. Hence, deleting the *Cs* gene stops cellular respiration in all cell types, making it impossible to examine the eCS function. The second *Cs* gene (hereafter referred to *eCs*) encodes eCS in mice [7] (Figure 1a). CS and eCS proteins (GenBank accession nos. NP_080720.1 [8] and NP_082221.2 [9]) share high similarity (91.4%). An eCS-specific sequence is located between amino acids 216 and 229 (IYRNLYREDRNIEA), and this was used for the production of an eCS-specific polyclonal antibody (polyAb) [7]. In addition, *Cs* mRNA was expressed in all organs, whereas *eCs* mRNA was dominantly expressed in the testis. Moreover, eCS was detected in the epididymal sperm. Injection of *Cs* mRNA as well as *eCs* mRNA induces Ca^2+^ oscillation in mouse eggs [7], suggesting that this common enzymatic activity is involved in inducing Ca^2+^ oscillation.

Immediately after sperm fusion, substances known as sperm factors enter the egg cytoplasm and evoke repetitive Ca^2+^ increases (Ca^2+^ oscillation), leading to a meiotic resumption in eggs [10]. Sperm-derived phospholipase C zeta 1 (PLCz1) is an enzyme that induces Ca^2+^ oscillation in mammalian eggs [10]. In the newt, CS, but not PLCz1, induces Ca^2+^ rise, leading to egg activation [11]. In addition, injection of *eCs* mRNA induces Ca^2+^ oscillation in mouse eggs in the absence of sperm extracts or PLCz1 [7]. PLCz1 expression and localization in the sperm are also comparable between *eCs*-deficient (*eCs*-KO) and wild-type (WT) sperm [7]. *eCs*-KO male mice produce an equivalent number of pups to WT male mice, but their fertility drops when they are over 6 months old [7]. Moreover, eCS triggers the first spike in Ca^2+^ oscillation in the presence of PLCz1 [7], implying that these two sperm factors work independently, at least in mice (Figure 1b). Sperm from *Plcz1*-deficient male mice failed to trigger Ca^2+^ oscillation in eggs, causing polyspermy [12,13]. Decreased *PLCz1* expression has also been observed in patients with infertility [14]. This study suggests that the failure of Ca^2+^ oscillation is closely related to polyspermy.

Polyspermy is often observed when in vitro fertilization (IVF) and intracytoplasmic sperm injection (ICSI) are performed using sperm from infertile patients [14]. Immature eggs also exhibit a higher percentage of polyspermy because of an incomplete ability to block polyspermy [15]. Regardless of problems on either the male or female side, polyspermy is believed to be one of the causes of reproductive failure in humans [14,16].

Our interest is the fate of eggs fused with multiple sperm and whether mammalian eggs have a mechanism of cytoplasmic sperm selection. In other words, given that mammalian sperm induces slow egg activation, how do such sperm and eggs behave? To address this, we focused on eggs fused with *eCs*-KO sperm.

## 2. Results

### 2.1. Subcellular Localization of CS and eCS Proteins

The expression vectors were transfected into human embryonic kidney 293T (HEK293T) cells to examine the intracellular localization of CS and eCS proteins (Figure 1b and Appendix A). These proteins were expressed as proteins fused to enhanced green fluorescent proteins (EGFP). CS was localized to the mitochondria and excluded from the nucleus, whereas eCS was widely localized in the cytoplasm and the nucleus. This result suggests that their localization is distinct and that eCS is localized outside the mitochondria (Figure 1c).

### 2.2. Sperm Concentration and Ca^2+^ Oscillation Pattern

We examined the occurrence of polyspermy in *eCs*-KO sperm and WT eggs to explore the possibility that delayed initiation of Ca^2+^ oscillations could cause polyspermy. We first estimated the initiation time of Ca^2+^ oscillation in WT eggs fused with *eCs*-KO sperm by increasing the sperm concentration (Figure 2a). The egg plasma membrane was exposed by removing the cumulus cells and an extracellular coat called the zona pellucida (zona). Zona pellucida-free (zona-free) eggs were loaded with Oregon Green 488 BAPTA1-AM and incubated with sperm. To examine the influence of the increased number of *eCs*-KO sperm, eggs were incubated with 30 sperm/µL (WT), 30 sperm/µL (*eCs*-KO), or 150 sperm/µL (*eCs*-KO) (Figure 2b).

The percentage of Ca^2+^ oscillation-induced eggs was strikingly reduced in eggs incubated with 30 sperm/µL of *eCs*-KO sperm compared to those incubated with WT sperm (Figure 2c,d). However, when the eggs were incubated with 150 sperm/µL of *eCs*-KO sperm, the percentage of Ca^2+^ oscillation-induced eggs tended to increase compared to those incubated with 30 sperm/µL of *eCs*-KO sperm (Figure 2c,d).

### 2.3. Sperm Concentration and Initiation of Ca^2+^ Oscillation

We further compared four parameters (i.e., amplitude, interval, frequency, and time of the first spike) in the Ca^2+^ oscillation patterns (Figure 3a). The amplitudes were comparable among the four parameters (Figure 3b). The time interval for oscillation was prolonged in eggs incubated with the 30 sperm/µL of *eCs*-KO sperm compared to that for eggs incubated with the WT sperm (479.0 ± 79.5 s vs. 260.0 ± 51.2 s; *p* = 0.025) (Figure 3c). In contrast, the interval tended to be shorter for eggs incubated with 150 sperm/µL of *eCs*-KO sperm (397.3 ± 106.4 s) (Figure 3c). The frequency of Ca^2+^ oscillation was also reduced in eggs incubated with 30 sperm/µL of *eCs*-KO sperm compared to those incubated with WT sperm (2.3 ± 0.9 vs. 5.8 ± 0.8; *p* = 0.026) (Figure 3d). The frequency increased in eggs incubated with 150 sperm/µL of *eCs*-KO sperm (3.8 ± 1.6).

The initiation of the first spike was also seen to be significantly retarded in eggs incubated with the 30 sperm/µL of *eCs*-KO sperm compared to those incubated with the WT sperm (907.1 ± 193.1 s vs. 421.6 ± 71.9 s; *p* = 0.032) (Figure 3e). In contrast, the initiation period was shorter in eggs incubated with 150 sperm/µL of *eCs*-KO sperm (529.0 ± 131.9 s).

Under normal conditions, multiple sperm cannot fuse with ZP-intact eggs and even with ZP-free eggs [17], and Ca^2+^ oscillation occurs in a membrane-fusion-mediated manner [10]. Unexpectedly, even when eggs were incubated with excess *eCs*-KO sperm, the Ca^2+^ oscillation pattern improved, implying that polyspermy might have occurred.

### 2.4. Multiple Fusion of eCs-KO Sperm with Zona-Free Eggs

To explore the occurrence of polyspermy under zona-free conditions, we examined the zona-free eggs fused with *eCs*-KO sperm (Figure 3a). As reported recently [7], the fertility of >6-month-old *eCs*-KO males strikingly decreased because of the low ability to induce Ca^2+^ oscillation in their sperm. Therefore, we collected sperm from 8–12-week-old and 6-month-old males and compared their fusion abilities. As exhibited in Figure 2 and Figure 3, the high concentration of *eCs*-KO sperm (150 sperm/µL) reversed the low ability of Ca^2+^ oscillation. When ZP-free eggs were incubated with *eCs*-KO sperm at 30 sperm/µL, the fusion rate was reduced (33.8 ± 3.2%; *p* < 0.001) (Figure 4b). In contrast, when the eggs were incubated with the sperm at 150 sperm/µL, they efficiently fused with the sperm, regardless of WT and *eCs*-KO sperm (Figure 4a,b).

Next, we examined the occurrence of multiple sperm fusions in eggs fused with *eCs*-KO sperm (Figure 4a,c). Expectedly, ZP-free eggs, fertilized with *eCs*-KO sperm, displayed the average number of 1.3 and 1.6 sperm fusions per egg (8–12-week-old and 6-month-old males, respectively), compared with that of 1.0 WT sperm fusion per egg, indicating that multiple *eCs*-KO sperm fused with an egg regardless of male age (Figure 4a,c). When ZP-free eggs were inseminated with *eCs*-KO sperm at 150 sperm/µL, the maximum number of fused sperm was two sperm per egg, and the percentage of ZP-free eggs fused with two sperm was 24.29% (8–12-week-old males) (Figure 4a,d).

Since sperm IZUMO1 controls sperm–egg fusion [18], its amount may affect the efficiency of sperm–egg fusion. Hence, we performed immunoblotting, but the expression of IZUMO1 protein was comparable between *eCs*-KO and WT sperm (Figure 4e and Appendix A).

This result reinforces the idea that polyspermy occurs under zona-free conditions due to insufficient inducers of Ca^2+^ oscillation.

### 2.5. Multiple Fusion of eCs-KO Sperm with Cumulus-Intact Eggs

The use of cumulus-intact eggs is suitable for predicting egg behavior in vivo. Next, we examined the occurrence of polyspermy under cumulus- and zona-intact (zona-intact) conditions. Similar to zona-free eggs, 47.6 ± 3.8% of zona-intact eggs fertilized with *eCs*-KO sperm also exhibited polyspermy compared with those fertilized with WT sperm (47.6 ± 3.8%; *p* < 0.001) (Figure 5 and Appendix A). Moreover, the levels of polyspermy were significantly higher in the sperm of 6-month-old males than in those of 8–12-week-old males (67.3 ± 3.6%; *p* = 0.001). Based on these results, we hypothesized that polyspermy could occur in vivo.

### 2.6. Two-Cell Embryos and Blastocysts from Eggs Fertilized with eCs-KO Sperm

To explore the fate of eggs fused with multiple *eCs*-KO sperm, we estimated the percentage of two-cell embryos and blastocysts (Figure 6a,b). Two-cell embryos were comparably formed from WT and *eCs*-KO sperm (63.4 ± 1.6 and 59.8 ± 3.6) (Figure 6c). Similarly, blastocysts were comparably formed from WT and *eCs*-KO sperm (61.0 ± 0.1 and 54.0 ± 7.9) (Figure 6d). From this result, we considered that polyspermy eggs could develop.

## 3. Discussion

In the present study, we examined the relationship between polyspermy and Ca^2+^ oscillation in mammals and subsequently explored the fate of polyspermic eggs. The elevated *eCs*-KO sperm concentration reversed the retarded Ca^2+^ oscillation initiation. eCS deficiency enhanced polyspermy in both the ZP-free and ZP-intact eggs (Figure 6). Subsequently, polyspermic eggs develop normally because one of the multiple sperm nuclei was selected in the egg cytoplasm. Our results suggest that mouse eggs develop normally under polyspermy conditions caused by retarded Ca^2+^ oscillation initiation due to cytoplasmic sperm selection.

### 3.1. Physiological Polyspermy

The union between the sperm and egg nuclei is necessary to mingle the genetic material as a diploid genome. Only one sperm is incorporated into an egg (monospermy); however, some animals utilize physiological polyspermy [5]. In contrast, only one sperm nucleus (the principal sperm nucleus) forms a zygotic nucleus with the egg nucleus in both fertilization types, implying the presence of cytoplasmic sperm selection in physiological polyspermy [5]. The principal sperm nucleus forms a larger aster than the remaining sperm (accessory sperm) that are degraded in the egg cytoplasm [5]. In chickens, 20–60 sperm are generally found within the egg cytoplasm, and this number is markedly higher than that of other polyspermic species; however, avian-specific events such as the degeneration of supernumerary sperm nuclei during early embryo development allow polyspermic eggs to develop normally [19].

### 3.2. Mechanisms of Physiological Polyspermy

*Plcz1*-KO sperm induce polyspermy in both mice [12,13] and patients with fertilization failure [14,20,21,22,23], suggesting that problems in Ca^2+^ signaling are related to polyspermy.

Microtubules spreading in egg cytoplasm are important for transporting organelles and other cytoplasmic components. Several mechanisms underlie the degradation of accessory sperm nuclei and centrosomes. In newt eggs, DNase activity in the egg cytoplasm participates in the selective degeneration of accessory sperm nuclei.

The ubiquitin–proteasome system degrades the components of the nuclei and centrosomes of the accessory sperm. Sperm mitochondria are also degraded by the ubiquitin–proteasome system. Accessory sperm nuclei, but not nuclei from the principal sperm and zygotes, are highly ubiquitinated. Several animals, including birds, reptiles, and most urodele amphibians, exhibit physiological polyspermy. However, the relationship between polyspermy and the ubiquitin–proteasome system remains unclear.

### 3.3. Pathological Polyspermy

Pathological polyspermy can occur for several reasons. The zona is the protective layer surrounding the eggs. If it is weakened or damaged, it may allow more than one sperm to penetrate [14]. Hyperactive motility or an excessive number of sperm can also increase the chances of polyspermy [14]. In some cases, assisted reproductive technologies, such as IVF or intracytoplasmic sperm injection, result in polyspermy [24].

Pathological polyspermy has been investigated as a byproduct to understand the mechanism of the fertilization system. Usually, a system against polyspermy (polyspermy block) suppresses the entry of additional sperm into the eggs. Therefore, polyspermy occurs when the polyspermy block is imperfect [6]. In pathological polyspermy, each additional sperm forms an aster that behaves as a centrosome [6]. Subsequently, multiple microtubule-organizing centers are formed and cause developmental problems. All sperm nuclei incorporated into the eggs form sperm pronuclei with well-developed asters [24]. Subsequently, polyspermic eggs undergo multipolar cleavage at the first cleavage stage, and embryonic development is stopped because of genomic mosaicism or aneuploidy.

In humans, mutations in *PLCZ1* genes cause male infertility and fertilization failure of intracytoplasmic sperm injection (ICSI) [14,20,21,22,23]. PLCZ1 expression and its subcellular pattern are significantly correlated with ICSI success rate. As reported previously [14,20,21,22,23], artificial oocyte activation (AOA) could rescue the lack of Ca^2+^ oscillation caused by mutations in *PLCZ1* genes. However, when patients with bi-allelic *PLCZ1* mutations were treated by ICSI with AOA, only one-fourth of them could have their own babies, indicating that PLCZ1 plays a role not only in fertilization, but also in embryonic development. In mouse eggs, PLCz1-induced Ca^2+^ oscillation is long-lasting, whereas that induced by eCS is short-lived [7]. Multiple spot applications of *eCs* mRNA or eCS proteins may be effective in the absence of PLCz1. If there is male infertility involving eCS, it would be in patients with normal *PLCZ1* expression and age-dependent loss of fertility.

Most cases of polyspermy result in failed pregnancies or spontaneous abortions. However, it is essential to note that not all cases of polyspermy lead to negative outcomes in mice and probably in humans.

## 4. Materials and Methods

### 4.1. Antibodies and Reagents

A rabbit polyAb against a synthetic peptide corresponding to the N-terminal amino acids (IYRNLYREDRNIEA) of mouse eCS, produced by Japan Lamb Ltd. (Hiroshima, Japan), was used for immunostaining and immunoblotting [7]. The rat anti-mouse IZUMO1 monoclonal antibody (mAb) required for immunoblotting was kindly provided by Dr. Ikawa (Osaka University, Japan). For immunoblotting, rabbit anti-CS polyAb (NE040/7S) was purchased from Nordic MUbio (Susteren, The Netherlands). Horseradish peroxidase (HRP)-conjugated secondary Abs (Sigma-Aldrich, St. Louis, MO, USA) was used for immunoblotting. Nuclei were counterstained with 4′,6-diamidino-2-phenylindole (DAPI) (WAKO Pure Chemical Industries, Osaka, Japan) or Hoechst 33342 (WAKO Pure Chemical Industries). Mitochondria were stained with 500 nM MitoTracker Red CMXRos (Invitrogen Corp., Waltham, MA, USA).

### 4.2. Transfection of Expression Vectors of CS and eCS Proteins

The cDNA fragments encoding the CS and eCS proteins were inserted into the HindIII and BamHI sites in the multiple cloning sites (MCS) of pcDNA3.1+C-eGFP (Genscript Biotechnology Co., Ltd., Nanjing, China) to express C-terminally enhanced green fluorescent protein (EGFP)-tagged proteins (CS and eCS) in mammalian cells. The expression vectors were transiently transfected into a cell line derived from the human embryonic kidney HEK293 cell line (HEK293T) using Lipofectamine 3000 (Thermo Fisher Scientific, Waltham, MA, USA). EGFP protein expression was observed after 48 h of incubation. The mitochondria were stained with 500 nM MitoTracker Red CMXRos. Nuclei were counterstained with DAPI.

### 4.3. eCs-KO Mice

As reported previously [7], mutant mice were generated from C57BL/6-derived embryonic stem cell clones by injecting blastocysts from C57BL/6 mice with genetically deleted *eCs* obtained from the knockout mouse project (KOMP) repository (an NCRR-NIH-supported strain suppository). C57BL/6J female and male mice were purchased from Japan SLC Inc. (Shizuoka, Japan). All the mice were housed under specific pathogen-free conditions. Food and water were provided ad libitum. All the animal experiments were approved by The Institutional Animal Care and Use Committee of the National Research Institute for Child Health and Development (experimental number A2004-004).

### 4.4. Measurement of Intracellular Ca^2+^ Concentration

Eggs were isolated from the oviducts of superovulated 8–12-week-old C57BL/6J female mice, and cumulus cells were removed from the eggs by treatment with hyaluronidase (300 μg/mL; Merck, Darmstadt, Germany) in Toyoda–Yokoyama–Hoshi (TYH) medium [19]. The ZP-intact eggs were then treated with collagenase (Wako Pure Chemical Industries) at a final concentration of 1 mg/mL in a 100-µL drop of TYH medium for 20 min at 37 °C [25]. ZP-free eggs were incubated in TYH medium containing the Ca^2+^-sensitive fluorescent dye Oregon green 488 BAPTA-1 AM (final concentration of 2 µM, Invitrogen) for 15 min at 37 °C in a CO_2_ incubator to monitor changes in intracellular Ca^2+^ concentration. The cells were then washed three times with TYH medium (5 min each wash). After washing, capacitated sperm (30 and 150 sperm/µL) were added to the TYH medium. Fluorescent images of the eggs were captured every 10 s using highly sensitive software (Yokogawa, Tokyo, Japan) equipped with a CCD camera (Andor Technology, Belfast, UK). Fluorescence intensity was measured using the Andor IQ imaging software (version 1.10.1) (Andor Technology). Each fluorescence image (F) was subtracted from the image before injection or from the image with the lowest fluorescence intensity (F0). The fluorescence intensity of individual eggs was measured within a user-selected region that covered most of the area of each egg, and the mean intensity over the same area for each image in a time series was automatically analyzed. Changes in fluorescence intensity were reported as the F/F0 ratio after sperm insemination, as previously described [7].

### 4.5. In vitro Fertilization

Eggs were collected from the oviductal ampulla region of superovulated 8- to 12-week-old C57BL/6J female mice 14 to 16 h after hCG injection and placed in a 100-µL drop of TYH medium covered with paraffin oil (Nacalai Tesque, Inc., Kyoto, Japan) equilibrated with 5% CO_2_ at 37 °C. The capacitation of sperm collected from the epididymis of 12–20-week-old male mice was induced by incubation in TYH medium for 90 min in an atmosphere of 5% CO_2_ at 37 °C before insemination. The female and male pronuclei in fertilized eggs were stained with DAPI and observed under an IX71 microscope (Olympus, Tokyo, Japan) after 3 and 6 h of incubation. In addition, two-cell embryos were observed under an IX71 microscope after 24 h of incubation.

After being isolated from the oviduct, the eggs were incubated with sperm from WT and *eCs*-KO males. At 24 h after incubation, the number of two-cell embryos was counted. At 3.5 days after the embryos were transferred to KSOM medium [26], the number of blastocysts was counted.

### 4.6. Immunoblotting

Sperm were collected from the epididymides of 12- to 20-week-old C57BL/6J and *eCs*-KO mice. The samples were lysed in Laemmli’s SDS sample buffer containing 2% SDS, 62.5 mM Tris-HCl (pH 6.8), 0.005% bromophenol blue, and 7% glycerol, boiled for 10 min at 95 °C, and resolved using SDS-PAGE and 12% acrylamide gels prior to immunoblotting. Detection of the proteins and primary Abs (0.1 µg/mL) of interest was performed by enzyme-linked color development with HRP-conjugated secondary Abs (0.01 µg/mL).

### 4.7. Statistical Analysis

Significant differences were calculated using Student’s *t*-test. Statistical significance was set at 0.05 (*p* < 0.05). Values are expressed as mean ± standard error (SE).

## 5. Conclusions

Similar to non-mammalian eggs, mouse eggs normally develop under polyspermy conditions caused by problematic Ca^2+^ oscillations due to cytoplasmic sperm selection.

## Figures and Tables

**Figure 1 ijms-25-00285-f001:**
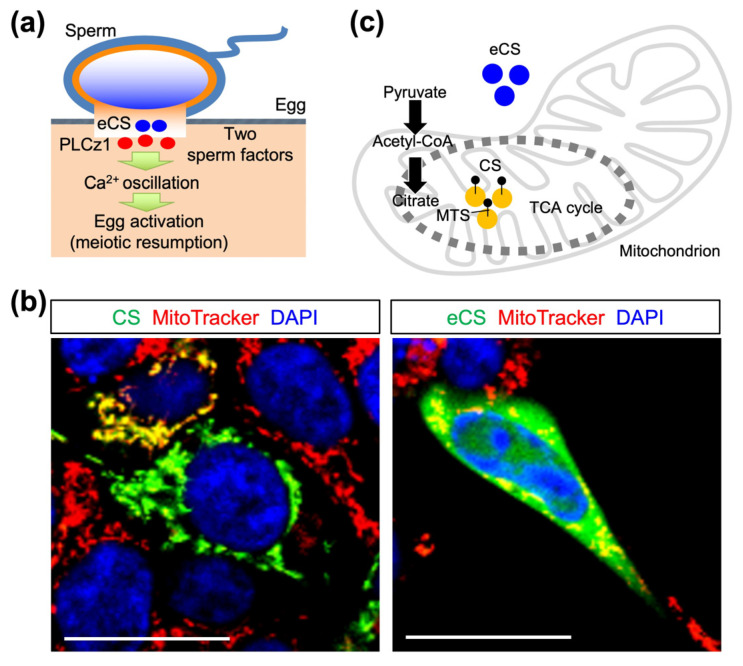
Subcellular localization of citrate synthase (CS) and extramitochondrial CS (eCS). (**a**) Sequential events of fertilization leading to Ca^2+^ oscillation. Blue circles: eCS; red circles: PLCz1. (**b**) Localization of CS and eCS proteins fused with EGFP. Their expression vectors were transfected into HEK293T cells. The cells were stained with DAPI and MitoTracker Red CMXRos. Green: CS and eCS fused with EGFP; red: mitochondria; blue: DAPI. Scale bars: 10 µm. (**c**) Two types of CS (intra- and extramitochondrial). MTS: mitochondrial targeting sequence; blue circles: eCS; yellow circles: CS; dotted circle: TCA cycle.

**Figure 2 ijms-25-00285-f002:**
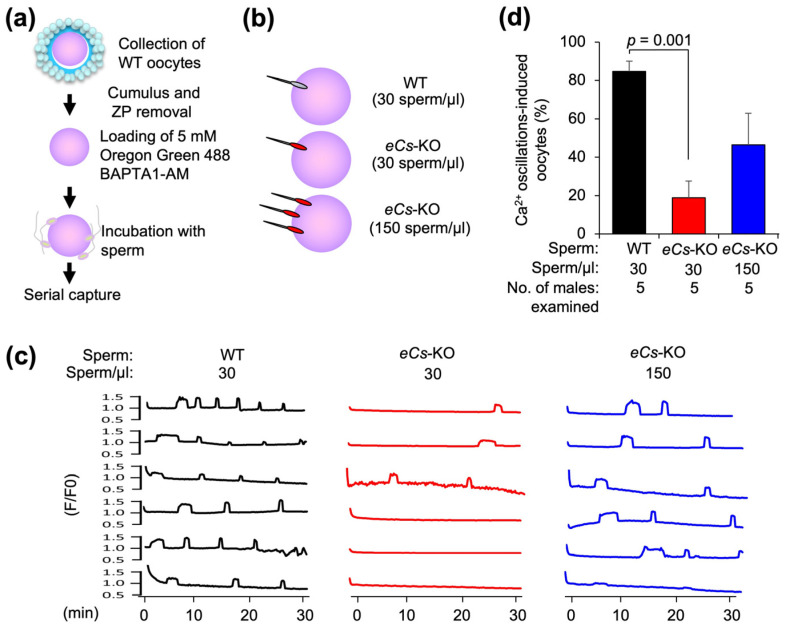
Induction of Ca^2+^ oscillation and sperm concentration. (**a**) Experimental flow for preparing eggs and monitoring of Ca^2+^ oscillation after sperm fusion. The egg plasma membrane was exposed by removing cumulus cells and the zona pellucida (zona). Zona-free eggs were loaded with Oregon Green 488 BAPTA1-AM and incubated with the sperm. (**b**) Incubation of eggs with the sperm at low and high concentrations. The eggs were incubated with 30 sperm/µL (WT), 30 sperm/µL (*eCs*-KO), and 150 sperm/µL (*eCs*-KO). (**c**) Sperm concentrations (30 and 150 sperm/µL) (3 experimental groups). Six patterns of Ca^2+^ oscillation were displayed in each experiment. (**d**) Percentage of Ca^2+^ oscillation-induced eggs. Values are expressed as mean ± SE.

**Figure 3 ijms-25-00285-f003:**
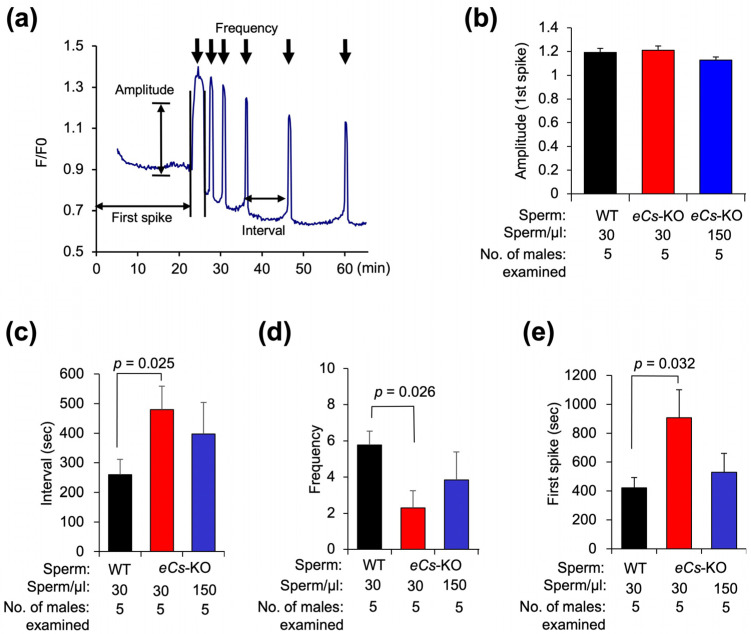
Correlation between Ca^2+^ oscillation pattern and concentration of *eCs*-KO sperm. (**a**) Evaluation of Ca^2+^ oscillation pattern. Four values (amplitude, interval, frequency, time until the first spike) were estimated for each group. The eggs were incubated with 30 sperm/µL (WT), 30 sperm/µL (*eCs*-KO), and 150 sperm/µL (*eCs*-KO). Arrows indicate Ca^2+^ rises. (**b**) Amplitude (first spike). (**c**) Interval (s). (**d**) Frequency (30 min). (**e**) Time until the first spike (s). Values are expressed as mean ± SE.

**Figure 4 ijms-25-00285-f004:**
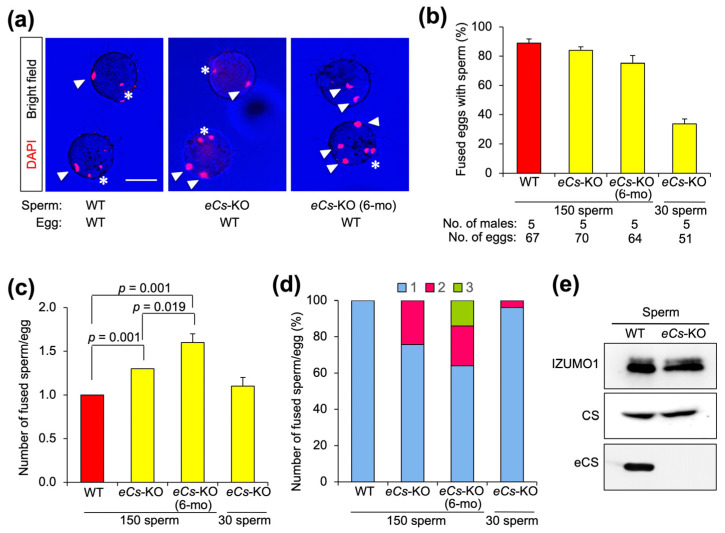
Polyspermy in zona-free eggs fused with *eCs*-KO sperm. (**a**) Polyspermic eggs. The eggs were incubated with 150 sperm/µL (WT), 150 sperm/µL (*eCs*-KO), and 150 sperm/µL (*eCs*-KO; 6 mo). Arrowheads: fused sperm heads; asterisks: egg chromosomes that resumed the MII-arrested cell cycle. Scale bar: 50 µm. (**b**) Number of fused eggs (%). The eggs were incubated with 150 sperm/µL (WT), 150 sperm/µL (*eCs*-KO), 150 sperm/µL (*eCs*-KO; 6 mo), and 30 sperm/µL (*eCs*-KO). Values are expressed as mean ± SE. (**c**) Number of fused sperm per egg. Values are expressed as mean ± SE. (**d**) Number of fused sperm per egg. Blue bars: a single sperm nucleus; pink bars: two sperm nuclei; green bar: three sperm nuclei. (**e**) Immunoblotting with anti-IZUMO1 mAb, and anti-CS, and anti-eCS polyAbs. The sperm were collected from the epididymis of WT and *eCs*-KO males.

**Figure 5 ijms-25-00285-f005:**
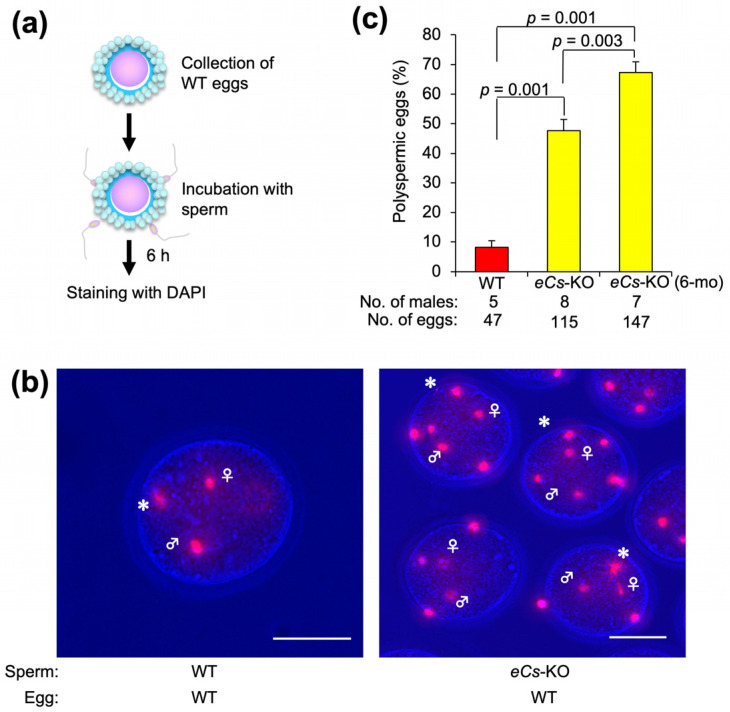
Polyspermy in zona-intact eggs fused with *eCs*-KO sperm. (**a**) Experimental flow for counting the number of zona-intact eggs fused with multiple sperm. After being isolated from the oviduct, the eggs were incubated with the sperm. The eggs were incubated with 150 sperm/µL (WT), 150 sperm/µL (*eCs*-KO), and 150 sperm/µL (*eCs*-KO; 6 mo). (**b**) Polyspermic eggs. Female and male nuclei were labelled. Asterisks: polar bodies. Scale bars: 50 µm. (**c**) Number of fused eggs (%). Values are expressed as mean ± SE. As reported previously [7], the litter size of *eCs*-KO males is comparable with that of WT males. Therefore, from our results, we suppose that one of multiple sperm nuclei may be selected in the egg cytoplasm.

**Figure 6 ijms-25-00285-f006:**
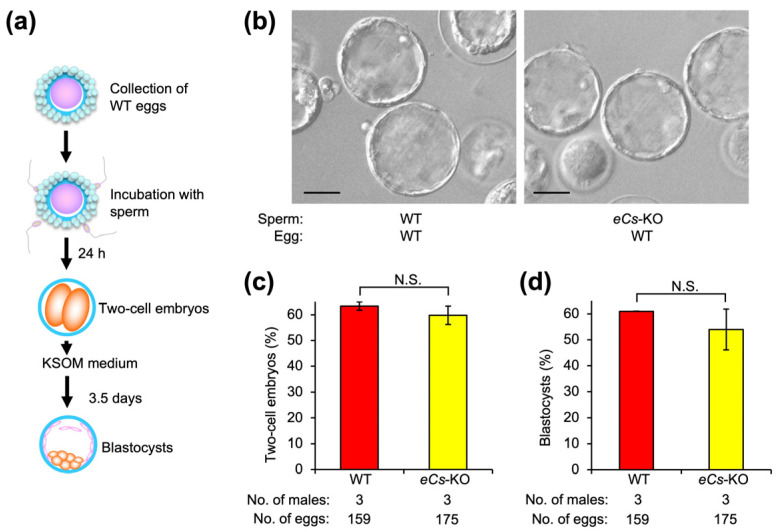
Two-cell embryos and blastocysts from eggs fertilized with *eCs*-KO sperm. (**a**) Experimental flow for counting the number of two-cell embryos and blastocysts. After being isolated from the oviduct, the eggs were incubated with sperm from WT and *eCs*-KO males. At 24 h after incubation, the number of two-cell embryos was counted. At 3.5 days after two-cell embryos were transferred to KSOM medium, the number of blastocysts was counted. (**b**) Blastocysts produced from *eCs*-KO sperm. Scale bars: 50 µm. (**c**) Formation of two-cell embryos (%). (**d**) Formation of blastocysts (%). N.S.: not significant. Values are expressed as mean ± SE.

## Data Availability

The data supporting the findings of this study are available from the corresponding author upon request.

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
