# Peer review of "Overdue Calcium Oscillation Causes Polyspermy but Possibly Permits Normal Development in Mouse Eggs"

_ijms, 2023, doi:10.3390/ijms25010285_

Round 1

Reviewer 1 Report

Comments and Suggestions for Authors

In this research article, the authors studied the role of extra-mitochondrial citrate synthase (eCs) in mammalian fertilization by use of the sperm derived from eCs-knockout mice. The authors demonstrated that fertilization of mice eggs with the eCs-KO sperm led to delay of Ca2+ oscillation and increase of the polyspermy rate. The authors argued that these eggs nonetheless grow normally in most cases, and thereby suggested that the mouse egg might have a way to ‘select’ single sperm out of the multiple ones that entered the egg in a model similar to ‘physiological polyspermy’ which is known to occur only in the species such as birds and reptiles that have large eggs. Hence, if proved true, the authors’ claim may be a significant finding in the field of reproductive biology. However, it appears that the manuscript have some serious holes in building the arguments, which should be fixed in a formal way.

Specific points:

1. Insemination with eCs-KO sperm led to repression of the Ca2+ signal patterns in the fertilized eggs, as judged by the time lag, frequency, and amplitude. Supplementing the insemination media with dbcAMP de-repressed them back to the Ca2+ spike patters similar to those of the eggs fertilized with WT sperm (Fig. 3). However, the supplementary dbcAMP notably did not restore the percentage of the eggs displaying the Ca2+response at fertilization. The histogram bar of the eCs+dbcAMP is merely half the value to that in the WT sperm (Fig. 2D). In this context, it is curious to know whether dbcAMP alleviates the polyspermy rate that is increased (>45%) by the eCs-KO sperm, but such a result is absent in Fig. 5 C. Hence, we have no direct evidence that the delayed (or repressed) Ca2+ signaling in these eggs are the ‘cause’ of the increased polyspermy rate. Being a metabolic enzyme, eCs could do many different things related to energy supply, which is in parallel with Ca2+ signaling. In that sense, the title claiming the causal relationship between the Ca2+ signaling and polyspermy is an overstatement.

2. According to the presented data, >45% of zygotes produced by eCs-KO sperm are polyspermic (Fig. 5C). At the later stages (2 cell and blastocysts), about 50% of the embryos are morphologically normal, which was the basis of the authors’ argument that there is a phenomenon of ‘physiological polyspermy’ remedying the presence of multiple sperm in the fertilized mouse eggs. However, it appears again that there is no direct evidence supporting that claim. It is simply unknown whether those embryos that developed normally at later stages had actually started from polyspermic zygotes. Because the monospermic zygotes were close to 55%, these normally developing embryos may well be the ones that developed from monospermic zygotes. There is no demonstration that polyspermic zygotes actually became normally growing embryos. There is only a weak statistics with no analytic test supporting the idea. For this, the fate of individual polyspermic zygotes must be carefully followed. The authors demonstrated possession of the methodology, but it was not pursued. In the given circumstances with the lack of direct evidence, the second part of the title implying that physiological polyspermy was allowed to progress and led to normal development in mice egg is again a huge overstatement.

3. It seems that the Introduction is not self-explanatory when it describes gene(s) encoding CS and the difference between the mitochondrial and extra-mitochondrial variants. It must provide sufficient information so that the readers can understand the narration without having to read the cited papers. Related to that, the evidence provided to support the argument that eCs and PLC-z1 are ‘two sperm factors that work independently’ (lines 45-46)” is not convincing. The statement rather sounds like that PLC-z1 overrides the Ca2+ deficit caused by eCs. How can it be an evidence that eCs is independent of PLC-z1? In Line 49, patients should be specified perhaps as “patients with infertility” or something like that. These are just a few examples. Overall, the Introduction does not really usher the readers in to the precise research topic, and the intent of the study is often obscure. This rather short Introduction must be elaborated much more.

4. In Fig. 1, the authors apparently intended to show that CS is localized in mitochondria and eCS outside them, but the signals in panel B indicate that both CS-GFP and eCS-GFP are not co-localized with the fluorescent markers for mitochondria. Hence, it is only confusing because what it shows does not match what is said. The legends of this and all other figures seem to require precise description of what was done and what is shown. The method of microscopy (confocal vs. epifluorescence), etc. etc. should be specified more. In related to this, somehow isolated result utilizing a HEK293 cells transfected with the GFP fused with Cs and eCS makes less sense. The authors opted to demonstrate the subcellular localization of eCS in the heterologous cell system even though they have access to functional eCs and CS antibodies with proven specificity (Fig. 4e). They could have tried immunofluorescence (eCS and CS) together with MitoTracker in the mouse eggs or embryos, which are actually the cells of interest and are even bigger than HEK293 cells.                                                      

Comments on the Quality of English Language

None

Author Response

November 26, 2023

Manuscript ID: ijms-2719226

Title: Overdue calcium oscillation causes polyspermy but permits normal development in mouse eggs

Dear Editorial board member, IJMS,

Thank you for your e-mail (November 19, 2023) and the comments from reviewers concerning the above manuscript. We have improved our manuscript in response to the constructive criticism raised by the reviewer.

Our alterations and responses to the comments are shown below in detail, and our alterations are indicated in red in the revised manuscript.

Response to Review

Reviewer #1:

Comment 1: In this research article, the authors studied the role of extra-mitochondrial citrate synthase (eCs) in mammalian fertilization by use of the sperm derived from eCs-knockout mice. The authors demonstrated that fertilization of mice eggs with the eCs-KO sperm led to delay of Ca2+ oscillation and increase of the polyspermy rate. The authors argued that these eggs nonetheless grow normally in most cases, and thereby suggested that the mouse egg might have a way to ‘select’ single sperm out of the multiple ones that entered the egg in a model similar to ‘physiological polyspermy’ which is known to occur only in the species such as birds and reptiles that have large eggs. Hence, if proved true, the authors’ claim may be a significant finding in the field of reproductive biology. However, it appears that the manuscript have some serious holes inbuilding the arguments, which should be fixed in a formal way.

Response: First of all, we greatly appreciate your thoughtful review of our manuscript.

Comment 2: Insemination with eCs-KO sperm led to repression of the Ca2+ signal patterns in the fertilized eggs, as judged by the time lag, frequency, and amplitude. Supplementing the insemination media with dbcAMP de-repressed them back to the Ca2+spike patters similar to those of the eggs fertilized with WT sperm (Fig. 3). However, the supplementary dbcAMP notably did not restore the percentage of the eggs displaying the Ca2+ response at fertilization. The histogram bar of the eCs+dbcAMP is merely half the value to that in the WT sperm (Fig. 2D). In this context, it is curious to know whether dbcAMP alleviates the polyspermy rate that is increased (>45%) by the eCs-KO sperm, but such a result is absent in Fig. 5 C. Hence, we have no direct evidence that the delayed (or repressed) Ca2+ signaling in these eggs are the ‘cause’ of the increased polyspermy rate. Being a metabolic enzyme, eCs could do many different things related to energy supply, which is in parallel with Ca2+ signaling. In that sense, the title claiming the causal relationship between the Ca2+ signaling and polyspermy is an overstatement.

Response: Thank you for your valuable comment. As the reviewer pointed out, the effect of dbcAMP was incomplete. The dbcAMP treatment does not completely mimic the intracellular system, because its effects are limited. Our results suggest that its treatment is at least effective in the induction of Ca2+ oscillation. As reported previously (reference No. 7), injection of Cs mRNA as well as eCs mRNA induces Ca2+ oscillation in the absence of sperm extracts or PLCZ1, indicating that these common enzymatic activity contributes to the induction of Ca2+ oscillation. Furthermore, Plcz1-KO sperm induce polyspermy in both mice and patients with fertilization failure, suggesting that problems in Ca2+ signaling are related to polyspermy. We have added the sentences in the discussion section (lines 259 – 263 and corresponding references).

Comment 3: According to the presented data, >45% of zygotes produced by eCs-KO sperm are polyspermic (Fig. 5C). At the later stages (2 cell and blastocysts), about 50% of the embryos are morphologically normal, which was the basis of the authors’ argument that there is a phenomenon of ‘physiological polyspermy’ remedying the presence of multiple sperm in the fertilized mouse eggs. However, it appears again that there is no direct evidence supporting that claim. It is simply unknown whether those embryos that developed normally at later stages had actually started from polyspermic zygotes. Because the monospermic zygotes were close to 55%, these normally developing embryos may well be the ones that developed from monospermic zygotes. There is no demonstration that polyspermic zygotes actually became normally growing embryos. There is only a weak statistics with no analytic test supporting the idea. For this, the fate of individual polyspermic zygotes must be carefully followed. The authors demonstrated possession of the methodology, but it was not pursued. In the given circumstances with the lack of direct evidence, the second part of the title implying that physiological polyspermy was allowed to progress and led to normal development in mice egg is again a huge overstatement.

Response: Thank you for your useful comment. Accordingly, we have not been able to obtain data on the normal development of polyspermic eggs. As reported previously (reference No. 7; Figure 3c), eCs-KO male mice less than 6 months old (used in this study) produce an equivalent number of pups to wild-type (WT) male mice. From this result, we assumed that polyspermic eggs developed normally. We have added sentences regarding this point in the introduction section (lines 57 – 57). Since there are no direct evidence, the title and abstract have been revised (lines 1 and 21).

Comment 4: It seems that the Introduction is not self-explanatory when it describes gene(s) encoding CS and the difference between the mitochondrial and extra-mitochondrial variants. It must provide sufficient information so that the readers can understand the narration without having to read the cited papers. Related to that, the evidence provided to support the argument that eCs and PLC-z1 are ‘two sperm factors that work independently’ (lines 45-46)” is not convincing. The statement rather sounds like that PLC-z1 overrides the Ca2+ deficit caused by eCs. How can it be an evidence that eCs is independent of PLC-z1? In Line 49, patients should be specified perhaps as “patients with infertility” or something like that. These are just a few examples. Overall, the Introduction does not really usher the readers in to the precise research topic, and the intent of the study is often obscure. This rather short Introduction must be elaborated much more.

Response: Thank you for your valuable comment. CS and eCS proteins (GenBank accession no. NP_080720.1 and NP_082221.2) share high similarity (91.4%). However, eCS lacks the mitochondrial targeting sequence, which is important for mitochondrial localization. An eCS-specific sequence is located between amino acids 216 and 229 (IYRNLYREDRNIEA), and this was used for the production of an eCS-specific polyclonal antibody. In addition, Cs mRNA was expressed in all organs, whereas eCs mRNA was dominantly expressed in the testis. Moreover, eCS was detected in the epididymal sperm (Figure 1). Furthermore, injection of Cs mRNA as well as eCs mRNA induces Ca2+ oscillation in mouse eggs, suggesting that these common enzymatic activity is involved in its induction. According to the reviewer’s comment, we have added sentences concerning this point in the introduction section (lines 41 – 48).

Figure 1. Subcellular localization of eCS and CS in the sperm. (a) Localization of eCS and mitochondria. (b) Localization of CS and mitochondria. Insets were enlarged. Sperm were fixed and stained with an anti-eCS and anti-CS polyclonal antibodies (red) and MitoTracker Green FM (green). Scale bar, 10 mm. This figure was modified from reference No. 7.

As the reviewer mentioned, we have considered that eCs and PLCz1 work independently. In the newt, CS, but not PLCz1, induces Ca2+ rise, leading to egg activation. Injection of eCs mRNA induces Ca2+ oscillation in mouse eggs without sperm extracts or PLCz1, implying that eCS and PLCz1 work independently. We have added sentences regarding this point in the introduction section (lines 52 – 57).

Otherwise, eCs-KO mice show hair graying (Figure 2a, b). We isolated and cultured melanocytes from eCs-KO mice to determine the cause of their gray coat color (Figure 2c, d). Subsequently, we found that Ca2+ oscillation is involved in melanocyte formation. Melanocytes isolated from eCs-KO mice did not undergo Ca2+ oscillation. Since PLCz1 is not expressed in melanocytes, eCS and PLCz1 are thought to work independently. The data regarding melanocytes will be presented as the next paper.

Figure 2. Calcium oscillation is required for the transition from lysosomes to melanosomes (a) Hair graying in eCs-/- (KO) and wild-type (WT) mice. (b) Enlarged hair from KO, WT (C57BL/6J), and WT (ICR) mice. (c) Ca2+ oscillation in WT and KO melanocytes. Melanocytes were isolated from newborn mice and cultured. (d) Electron-microscopic images of WT and KO melanocytes. Percentage of pigmented melanocytes of WT and KO mice. Scale bars: 1 mm. Values are expressed as mean ± standard error.

As the reviewer pointed out, we specified the word “patients” as “patients with infertility” (line 61).

Comment 5: In Fig. 1, the authors apparently intended to show that CS is localized in mitochondria and eCS outside them, but the signals in panel B indicate that both CS-GFP and eCS-GFP are not co-localized with the fluorescent markers for mitochondria. Hence, it is only confusing because what it shows does not match what is said. The legends of this and all other figures seem to require precise description of what was done and what is shown. The method of microscopy (confocal vs. epifluorescence), etc. etc. should be specified more. In related to this, somehow isolated result utilizing a HEK293 cells transfected with the GFP fused with Cs and eCS makes less sense. The authors opted to demonstrate the subcellular localization of eCS in the heterologous cell system even though they have access to functional eCs and CS antibodies with proven specificity (Fig. 4e). They could have tried immunofluorescence (eCS and CS) together with MitoTracker in the mouse eggs or embryos, which are actually the cells of interest and are even bigger than HEK293 cells.

Response: Thank you for your valuable comment. eCS and CS proteins were separately localized in the sperm (Figure 1). eCS, but not CS, is localized in the sperm head. In contrast, CS, which should be localized in the mitochondria, is not localized in the midpiece, where the mitochondria are housed. Therefore, the sperm are highly specialized cells and cannot be considered like other types of cells. In addition, mature eggs and embryos do not express eCS. Therefore, we introduced both expression vectors into commonly used HEK293 cells.

Figure 3. Distribution of PLCz1 in WT and KO sperm. (a) Immunostaining of PLCz1 in sperm. Sperm were stained with an anti-PLCz1 polyclonal antibody (red). Scale bar, 10 μm. The fluorescence intensity profiles of PLCz1-stained areas in sperm head shown in a. Fluorescence intensities were measure along with white dotted lines. (b) Immunoblotting of testis protein extracts. Blots were probed using antibodies against sperm proteins, as indicated. (c) Fertility of WT and KO males. Male mice were mated with WT and KO females. In each group, 9 males were tested. Older WT and eCs-KO male mice (> 6-month-old) were mated with eCs-KO females (8–10-week-old). In total, 10 and 20 older males were tested, respectively. This figure was modified from reference No. 7.

My coauthors and I think that the revised manuscript has been fundamentally improved and that it includes the contents requested by the reviewers and editorial team.

Thank you for your time and consideration.

Sincerely yours,

Kenji Miyado, PhD

Department of Reproductive Biology, National Research Institute for Child Health and Development, 2-10-1 Okura, Setagaya, Tokyo 157-8535, Japan

Phone: +81-3-5494-7047

Fax: +81-3-5494-7048

Reviewer 2 Report

Comments and Suggestions for Authors

Generally, I think that the authors have thoroughly investigated the role of eCS in the current manuscript, but seem to have not investigated a major player which is PLCzeta.

This is made more prudent by recent PLCZ1 KO mice studies, that all indicate a high level of polyspermy in KO mice, but also seem to exhibit some slight (and delayed) release of calcium (generally abnormal profiles but enough to apparently result in successful fertilisation and embryogenesis. 

Given that the main contention of the authors is the role of eCS in this process, as well as the assertions previously made by the authors that both PLCZ1 and eCS work in conjunction, it would have also been prudent to examine PLCZ1 in eCS-KO sperm. Are levels of PLCZ1 comparable to WT? This would add more strength to their argument and the story seems incomplete without this data - particularly since the exact relationship between eCS and PLCZ1 (and both their roles are oocyte activation) are unclear currently.

It would also be useful for authors to comment/investigate the role of eCS in PLCZ1 KO mice as there is an interesting phenotype whereby ICSI of PLCZ1 KO sperm does not result in calcium release, while this does happen in a minority of cases with IVF using such sperm. The accompanying phenotype is high levels of polyspermy and this seems to match with the phenotype of the eCS-KO mice. 

Author Response

Response to Review

Reviewer #2:

Comment 1: Generally, I think that the authors have thoroughly investigated the role of eCS in the current manuscript, but seem to have not investigated a major player which is PLCzeta.

Response: First of all, we greatly appreciate your thoughtful review of our manuscript.

Comment 2: This is made more prudent by recent PLCZ1 KO mice studies, that all indicate a high level of polyspermy in KO mice, but also seem to exhibit some slight (and delayed) release of calcium (generally abnormal profiles but enough to apparently result in successful fertilisation and embryogenesis). Given that the main contention of the authors is the role of eCS in this process, as well as the assertions previously made by the authors that both PLCZ1 and eCS work in conjunction, it would have also been prudent to examine PLCZ1 in eCS-KO sperm. Are levels of PLCZ1 comparable to WT? This would add more strength to their argument and the story seems incomplete without this data - particularly since the exact relationship between eCS and PLCZ1 (and both their roles are oocyte activation) are unclear currently.

Response: Thank you for your useful comment. As reported previously (reference No. 7; Figure 1a, b), PLCz1 expression and localization in the sperm did not differ between WT and KO sperm. We have added sentences concerning this point in the introduction section (lines 54 – 56).

Figure 1. Distribution of PLCz1 in WT and KO sperm. (a) Immunostaining of PLCz1 in sperm. Sperm were stained with an anti-PLCz1 polyclonal antibody (red). Scale bar, 10 μm. The fluorescence intensity profiles of PLCz1-stained areas in sperm head shown in a. Fluorescence intensities were measure along with white dotted lines. (b) Immunoblotting of testis protein extracts. Blots were probed using antibodies against sperm proteins, as indicated. (c) Fertility of WT and KO males. Male mice were mated with WT and KO females. In each group, 9 males were tested. Older WT and eCs-KO male mice (> 6-month-old) were mated with eCs-KO females (8–10-week-old). In total, 10 and 20 older males were tested, respectively. This figure was modified from reference No. 7.

Otherwise, eCs-KO mice show hair graying (Figure 2a, b). We isolated and cultured melanocytes from eCs-KO mice to determine the cause of their gray coat color (Figure 2c, d). Subsequently, we found that Ca2+ oscillation is involved in melanocyte formation. Melanocytes isolated from eCs-KO mice did not undergo Ca2+ oscillation. Since PLCz1 is not expressed in melanocytes, eCS and PLCz1 are thought to work independently. The data regarding melanocytes will be presented as the next paper.

Figure 2. Calcium oscillation is required for the transition from lysosomes to melanosomes (a) Hair graying in eCs-/- (KO) and wild-type (WT) mice. (b) Enlarged hair from KO, WT (C57BL/6J), and WT (ICR) mice. (c) Ca2+ oscillation in WT and KO melanocytes. Melanocytes were isolated from newborn mice and cultured. (d) Electron-microscopic images of WT and KO melanocytes. Percentage of pigmented melanocytes of WT and KO mice. Scale bars: 1 mm. Values are expressed as mean ± standard error.

Comment 3: It would also be useful for authors to comment/investigate the role of eCS in PLCZ1 KO mice as there is an interesting phenotype whereby ICSI of PLCZ1 KO sperm does not result in calcium release, while this does happen in a minority of cases with IVF using such sperm. The accompanying phenotype is high levels of polyspermy and this seems to match with the phenotype of the eCS-KO mice.

Response: Thank you for your valuable comment. As the reviewer mentioned, mutations in PLCZ1 genes cause male infertility and fertilization failure of intracytoplasmic sperm injection (ICSI). Therefore, PLCZ1 expression and its subcellular pattern are significantly correlated with ICSI success rate.

As reported previously, artificial oocyte activation (AOA) could rescue the lack of Ca2+ oscillation caused by mutations in PLCZ1 gene. However, when patients with bi-allelic PLCZ1 mutations were treated by ICSI with AOA, only one fourth of them could have their own babies, indicating that PLCZ1 plays a role not only in fertilization, but also in embryonic development. PLCz1-induced Ca2+ oscillation is long-lasting, whereas eCS-induced one is short-lived. Multiple spot applications of eCs mRNA may be effective in the absence of PLCz1. If there is male infertility involving eCS, it would be in patients with normal PLCZ1expression and age-dependent loss of fertility.

Accordingly, we have inserted sentences concerning this point in the discussion section (lines 290 – 300 and corresponding references).

My coauthors and I think that the revised manuscript has been fundamentally improved and that it includes the contents requested by the reviewers and editorial team.

Thank you for your time and consideration.

Sincerely yours,

Kenji Miyado, PhD

Department of Reproductive Biology, National Research Institute for Child Health and Development, 2-10-1 Okura, Setagaya, Tokyo 157-8535, Japan

Phone: +81-3-5494-7047

Fax: +81-3-5494-7048

Round 2

Reviewer 1 Report

Comments and Suggestions for Authors

The authors' revision has improved the manuscript. 

Comments on the Quality of English Language

None

Author Response

Comment: The authors' revision has improved the manuscript.

Response: We greatly appreciate your comment.

Reviewer 2 Report

Comments and Suggestions for Authors

The authors have revised significant aspects of the manuscript which is now improved.

However, the authors have yet to completely explain the significance of their results in the discussion. The utilisation of the dbcAMP adds a layer of confusion in the entire study, and I was hoping that perhaps the authors would be able to relate this information with PLCz as previous studies indicate that both proteins are required to initiate the first calcium peak. 

However, I do not see any particular reasoning for using dbcAMP beyond artificially inducing calcium oscillations. Why was this necessary if eCS is involved predominantly in only the first calcium peak? The authors should also present data obtained without the use of dbcAMP as a control, irrespective of whether previous studies have examined this or not as this should be viewed in the context of the current study.

Is the polyspermy data obtained from oocytes/embryos treated with dbcAMP or without? This would be important as polyspermy seems to be linked to occurence or not of calcium oscillations, not necessarily just the first peak. 

The schematic presented by the authors in figure 7 is immensely speculative, with no data really presented in this study or previous ones to suggest its veracity. The only way this schematic can be presented and supported is by titrated co-injections of PLCZ1/eCS. I would suggest to remove this schematic as it is very misleading with no supporting data.

For the development aspect of polyspermic embryos - were these in the absence or presence of dbcAMP? What was the fate of non-polyspermic embryos? 

Author Response

Comment 1: The authors have revised significant aspects of the manuscript which is now improved.

Response: First of all, we greatly appreciate your thoughtful review of our manuscript.

Comment 2: However, the authors have yet to completely explain the significance of their results in the discussion. The utilisation of the dbcAMP adds a layer of confusion in the entire study, and I was hoping that perhaps the authors would be able to relate this information with PLCz as previous studies indicate that both proteins are required to initiate the first calcium peak. However, I do not see any particular reasoning for using dbcAMP beyond artificially inducing calcium oscillations. Why was this necessary if eCS is involved predominantly in only the first calcium peak? The authors should also present data obtained without the use of dbcAMP as a control, irrespective of whether previous studies have examined this or not as this should be viewed in the context of the current study. Is the polyspermy data obtained from oocytes/embryos treated with dbcAMP or without? This would be important as polyspermy seems to be linked to occurrence or not of calcium oscillations, not necessarily just the first peak.

Response: Thank you for your valuable comment. We apologize for confusion caused by the use of dbcAMP. Accordingly, we have removed all experiments using dbcAMP and improved Figures 2 and 3, and corresponding figure legends, and removed references concerning dbcAMP. We did not use dbcAMP in polyspermy experiments.

Comment 3: The schematic presented by the authors in figure 7 is immensely speculative, with nodata really presented in this study or previous ones to suggest its veracity. The only way this schematic can be presented and supported is by titrated co-injections of PLCZ1/eCS. I would suggest to remove this schematic as it is very misleading with no supporting data.

Response: Accordingly, we have removed Figure 7.

Comment 4: For the development aspect of polyspermic embryos - were these in the absence or presence of dbcAMP? What was the fate of non-polyspermic embryos?

Response: Thank you for your useful comment. We did not use dbcAMP in polyspermy experiments. We assume that non-polyspermic eggs are delayed or fail in transiting to two cell-staged embryos.

Round 3

Reviewer 2 Report

Comments and Suggestions for Authors

Manuscript is much improved.